# Research on Enterprise Interactive Innovation Balance Decision in Green Manufacturing Innovation Ecosystem

**Hao Qin, Hua Zou * and Huimin Ji**

School of Management, Shenyang University of Technology, Shenyang 110870, China;
qh_2020@smail.sut.edu.cn (H.Q.); jihm272@smail.sut.edu.cn (H.J.)
* Correspondence: suo-2001@163.com

**Abstract:** The green manufacturing innovation ecosystem provides a critical pathway for the interactive innovation balance between exploratory and exploitative technology innovation for green enterprise manufacturing. Under finite rationality, we construct a three-party evolutionary game model and its dynamic replication equations among the enterprise, scientific research, and support layers. We numerically simulate the decision-making behavior of the three parties with the BYD case study to analyze the influence of each parameter on the evolutionary outcome. The results show that the maturity and innovation degree of green manufacturing exploratory and exploitative innovation technologies can effectively measure the degree of innovation balance of the enterprise. Strengthening the scientific research layer to adopt green manufacturing technology innovation research and development for the enterprise and reducing the "conflict" will promote the enterprise to engage in exploratory innovation, which in turn will encourage the decision of enterprise interactive innovation balance. The support layer actively searches for information and supports the green manufacturing innovation ecosystem with information, funding, and other innovation resources, thus facilitating enterprises to engage in interactive innovation balance.

**Keywords:** green manufacturing; innovation ecosystem; innovation balance; sustainability; evolutionary game





## 1. Introduction

Green manufacturing is an advanced manufacturing model with the development of ecological civilization. Since 2016, China's green manufacturing system has steadily advanced; green products, green factories, green parks, and green supply chains have developed in tandem, and the scale of the green manufacturing industry has gradually grown, effectively promoting the green and low-carbon transformation of industry and contributing to the realization of the vision of carbon peaking and carbon neutral goals [1]. With the accelerated evolution of the new "green" technological revolution, green manufacturing has become the key to high-quality economic development. Technological innovation is the driving force of the new revolution to generate new development momentum. In the new era of innovation-led development, where the competition for green manufacturing is becoming increasingly fierce, the demand for green manufacturing products and services in emerging markets is diversified. Sustainable research and development (R&D) of new technologies, products, and services is the inevitable path for enterprises to enhance their competitive advantages in green manufacturing. Under the current innovation environment, enterprises are no longer a single innovation subject but build and continuously optimize the innovation ecosystem to achieve access and utilization of cross-border resources [2]. The key to developing green manufacturing in enterprises lies in the sustainable innovation and application of major green vital technologies. Green manufacturing technology innovation requires a perfect and sound "ecosystem," including the whole process of green manufacturing innovation from the source R&D to the transformation

of scientific and technological achievements and the final marketization [3]. In reality, HP, BYD, and Gree and other enterprises have built a green manufacturing innovation ecosystem response to the enterprise collaboration with other innovative subjects for resource integration and technological innovation, and R&D to enhance green manufacturing competitiveness sustainably.

Suppose the enterprise continues to develop new green manufacturing technologies and products in collaboration with other innovation subjects, ignoring the optimization of existing green manufacturing skills, processes, and structures. In that case, it will likely result in an imbalance between exploratory and exploitative innovation. Excessive exploratory innovation will lead to large resource consumption, increased innovation cost, and the probability of failure, directly reducing the enterprise's revenue. Extreme exploitative innovation will hinder the development of the enterprise's cognitive ability in specific innovation areas, inhibit the enterprise's ability to anticipate forward-looking technology, and miss the opportunity to seize the innovation highland. In an environment of increased competition in a globalized market, enterprises have been able to maximize the utility of resources and overcome the "innovation trap" by enhancing their position as the mainstay of the national innovation system and balancing exploratory and exploitative innovation [4]. Therefore, for the green manufacturing innovation ecosystem, when the external environment is stable and changing, the enterprise should focus on exploitative innovation; When the external environment changes in a revolutionary way, the enterprise should focus on exploratory innovation. This interactive balance of alternating exploratory and exploitative innovation slows down the consumption of innovation resources for the enterprise. It is an optimal approach for green manufacturing innovation ecosystems in the development stage.

This paper addresses the following scientific questions: what is the mechanism law of collaborative participation of enterprises and other innovation agents in innovation equilibrium in the green manufacturing innovation ecosystem? How does the degree of R&D of green manufacturing technology innovation affect innovation agents' revenue when enterprises' innovation equilibrium evolves from a low equilibrium degree to a high equilibrium degree? What effects do the strategic behaviors of innovation agents have on the innovation equilibrium of enterprises? Further, the research work of this paper is as follows. First, based on the green manufacturing innovation ecosystem, we innovatively divide innovation subjects into core enterprises, scientific research layer, and support layer, and examine how innovation subjects make decisions to maximize benefits in the dynamic evolution of interactive balanced exploratory and exploitative innovation, as well as the impact of the game behaviors of scientific research layer and support layer on the balanced development of enterprise innovation. Second, we construct an evolutionary game payment matrix for innovation subjects, consider the effects of technological maturity and degree of exploratory innovation on innovation benefits and costs, and explore the impact of different innovation balance degrees on the equilibrium solutions of the game for innovation subjects. Third, combining numerical simulation with BYD enterprise case study, we investigate the influence of different changes of multiple variables, such as the degree of technological innovation effort of the scientific research layer, the "conflict" between the scientific research layer and enterprise, and the perceived benefit of the support layer, on the willingness of innovation subjects to cooperate and innovate.

The remaining parts of this paper are as follows. The second part is the theoretical basis and literature review. The third part describes the model construction. The fourth part is the numerical simulation. The fifth part is the conclusion and implications. The sixth part is the research gaps and outlook.

## 2. Theoretical Basis and Literature Review

### 2.1. Green Manufacturing Innovation Ecosystem

Green technology innovation, represented by clean production, environmental technology, and low-carbon technology, is a new engine for the manufacturing industry to

promote sustainable economic development [5]. Compared with traditional technological innovation, the complexity and uncertainty of green manufacturing technological innovation are higher, and the development of exploratory innovation is hindered [6].

The institutional and environmental influences on green manufacturing innovation in the Chinese context have been researched from multiple perspectives. In terms of macro factors such as government guidance and subsidies, Cumming, Rui et al. selected data information on Chinese enterprises and found that inequality in political capital directly affects the ability of enterprises to obtain bank loans through political channels, which in turn affects their likelihood to invest in innovation [7]; Guo, Guo et al. used panel data of Chinese manufacturing enterprises from 1998–2007 to empirically analyze the impact of government R&D programs on enterprises' innovation industries [8]; Liu, Du et al. used data of Chinese listed enterprises from 2010–2016 to examine government R&D subsidies as a primary policy tool for market failure and concluded that ex ante incentives have a higher impact on enterprises' innovation performance than ex post incentives [9]; Zhao, Xu et al. used empirical data of Chinese provinces to examine the impact effect of the formulation and deployment of national R&D subsidy policies significantly advancing national technological progress [10]. In terms of the influence of market environment factors, Fang, Lerner, et al. used a DID model to empirically analyze the impact of knowledge industry protection on innovation in China before and after the privatization of SOEs, concluding that IPR protection enhances firms' incentives to innovate and that private firms are more sensitive to this than SOEs [11]; Rong, Wu et al. used patent data of Chinese listed firms from 2002–2011 found that the presence of institutional investors promotes firm innovation [12]; Tian, Kou, et al. argued that venture capital plays a crucial role in fostering enterprise technological innovation and dissects it from two perspectives: equity background and investment strategy [13]; Zhang, Mohnen investigated whether innovation in Chinese manufacturing firms prolongs survival time and found that both R&D and product innovation increases the chances of firm survival [14].

In addition, scholars have also conducted research on green manufacturing in general contexts, mainly from the perspectives of green manufacturing development level, influencing factors, realization paths, and technology applications. For instance, Mao and Wang et al. pointed out that the core technology for enterprises to achieve green manufacturing is artificial intelligence [15]. Song and Yu et al. proposed a green innovation strategy, which refers to manufacturing enterprises' efforts to obtain a sustainable competitive advantage by carrying out green technological innovation to meet stakeholders' expectations while making strategic decisions [16]. Song and Lin found that the R&D of green technology innovation in the manufacturing industry requires the support of production factors such as capital, labor, and knowledge, and financial agglomeration provides the basis for achieving this condition [17]. Ying and Li et al. argued that the internal and external drivers of green manufacturing are mainly the internal enterprise environment, market environment, and institutional environment [18].

The synergistic effect of institutional innovation and technological innovation has significantly promoted the development of green manufacturing [19], while the innovation ecosystem emphasizes inter-subjective collaborative innovation to achieve value co-creation, typically characterized by synergistic symbiosis [20]. The concept of an innovation ecosystem can be traced back to Moore's "enterprise innovation ecosystem" from a business perspective in 1993 [21], which was later defined by Ander [22]. Nowadays, enterprises are more concerned with the static institutional analysis of factor composition and resource allocation when conducting green manufacturing and emphasize the dynamic evolution of the mechanism of action among innovation subjects. Meng and Li et al. concluded that green innovation is a crucial path for manufacturing enterprises to build a resource-saving and environment-friendly oriented innovation ecosystem through a single case analysis of a traditional manufacturing company—Iceberg Group [23]. Zeng and Xue et al. studied the green innovation ecosystem and pointed out that the innovation subjects mainly include core enterprises, upstream and downstream enterprises in the

green supply chain, competing enterprises, complementary enterprises, government, universities, research institutes, users, and information intermediaries, and the environmental elements mainly include market environment, policy environment, economic environment, cultural environment, scientific and technological environment, and natural environment, in which the innovation subjects and the innovation environment form a complex system of symbiotic competition and dynamic evolution through the flow of innovation elements [24]. Considering the limited rationality of enterprises and other innovation subjects in the cooperative innovation game, Su and Wei studied the stabilization strategy of tripartite participation of government, enterprise, and the public in green technology innovation through an evolutionary game model [25]; Lu and Cheng et al. studied the dynamic impact of government subsidies on manufacturers' green R&D through an evolutionary game model [26].

### 2.2. Exploratory and Exploitative Innovation

Exploratory innovation brings emerging market customer demand and future long-term revenue, while exploitative innovation brings stable short-term revenue [27]. Based on the organizational learning perspective, March first defined explorative learning and exploitative learning, emphasizing that exploration is an organizational activity characterized by search, change, experimentation, risk-taking, and experimentation, while exploitative organizational activity embodies optimization, selection, action, and efficiency [28]. On this basis, scholars have gradually combined exploration and exploitation with technological innovation and proposed exploratory and exploitative innovation [29]. Moreover, scholars have uncovered different clusters of research knowledge. For instance, Danneels argued that exploratory innovation is the act of developing new technologies to meet new customer needs, and exploitative innovation refers to the act of optimizing existing technologies to serve customers [30]; Wang further suggested that exploratory innovation is matching new customer and market needs to explore new market opportunities or new technological services for the organization, and exploitative innovation is to broaden the existing knowledge and skills of the organization and optimize the existing technology system to achieve production and service efficiency [31]. Regarding methods and contexts for the research of exploratory and exploitative innovation, Ngo and Bucic et al. empirically analyzed 150 Vietnamese enterprises as a sample, concluding that exploratory and exploitative innovation is the primary way in which technology perception and market perception enhance enterprise performance [32]. Duodu and Rowlinson explored the direct role of internal and external social capital on exploratory versus exploitative innovation and the indirect role of absorptive capacity based on a linkage and knowledge base perspective using a least squares approach [33].

### 2.3. Interactive Innovation Balance

Exploration and exploitation achieve innovative coexistence organically and coupled in the same subject to reach a state of balance and achieve matching efficiency and adaptation [34]. Interactive innovation balance reflects that exploratory and exploitative innovation are mutually reinforcing and dependent on each other. Zhang and Shen et al. argued that the balance strategy improves an enterprise's buffering ability to cope with innovation uncertainty and facilitates the acquisition of a long-term competitive advantage [35]. Using individuals engaged in innovation development as subjects, Simon and Tellier distinguished innovation streams into developmental and exploratory projects, concluding that learning processes in the dual balance of innovation streams arise first within projects and then between projects [36]. Lawrence and Tworoger et al. empirically analyzed the balance between exploratory and exploitative innovation by enterprise leaders. They found that leaders could demonstrate flexibility in balance-switching behaviors, effectively enhancing enterprise innovation performance [37]. The optimal innovation balance model differs when enterprises are at different life cycle stages. Burgelman proposed the intermittent innovation balance model, emphasizing that enterprises interactively explore and

exploit innovations at different stages and that both create ambivalence [38]. Rui and Luo studied the optimal innovation balance model for enterprises at different stages. They found that startup enterprises are suitable for interactive innovation balance, and when enterprises enter the growth stage, they need to change the innovation balance model to simultaneity equilibrium [39].

*2.4. Gaps in the Current Literature*

The research perspective of the current literature is usually a specific research field or a disciplinary perspective, which has been explored from local to overall, effectively promoting the development of exploratory and exploitative innovation theory of enterprises. However, the following gaps remain. First, the research subject is relatively single and needs a systematic perspective to track and analyze. Most of the literature focuses on enterprises alone but rarely incorporates the core innovation ecosystem of enterprises into the research scope and needs to include the influence of other innovation subjects on the balance of interactive innovation of enterprises. Second, it is more subjective and requires rigorous model arguments or mathematical derivations. The current literature on exploratory and exploitative innovation balance uses mainly qualitative methods such as questionnaires and case studies, ignoring the mathematical and theoretical connections between the two developments. Finally, most of them are based on static perspectives, and few pieces of literature have been sorted out from dynamic evolution and game perspectives, leading scholars to lack a dynamic and systematic understanding of the evolutionary process of the interactive innovation balance theory of exploratory and exploitative innovation.

## 3. Interactive Innovation Balance Structure

The green manufacturing innovation ecosystem contains multiple subjects and an innovation environment, as shown in Figure 1. The scientific research layer includes universities, institutes, and other subjects, which collaborate with enterprises to carry out R&D of green manufacturing technology innovation. The support layer includes innovation service subjects such as financial institutions and information intermediaries, which actively search for information from outside the system as well as collect feedback from users on the use of new products and then transmit practical information to enterprises, as well as make a financial investment for enterprises. The balance between exploratory and exploitative innovation in green manufacturing promotes the sustainable development of the green manufacturing innovation ecosystem, and the decision-making behavior of other innovation subjects has a direct impact on the enterprise's decisions.

When enterprises' exploratory and exploitative innovation are both at low levels, the innovation subjects in the green manufacturing innovation ecosystem are hostile and low-energy, and their overall gains are the lowest. The interactive innovation balance will evolve in three directions, as shown in Figure 2. First, when the enterprise evolves to a flat trajectory of high exploratory innovation capability and low exploitative innovation capability, the new products and services obtained by transforming the R&D results of exploratory innovation will generate high revenue. The scientific research layer will provide sufficient technological innovation results for enterprises. The support layer will actively provide feedback to enterprises on information such as customer needs in emerging markets. The R&D of new technology innovation in green manufacturing requires constant trial and error, and the probability of innovation failure is high. Therefore, when the enterprise focuses only on exploratory innovation, it must consume many resources and is difficult to benefit in the short term. In this scenario, the overall revenue of the green manufacturing innovation ecosystem fluctuates significantly, and it takes work to maintain a high level of momentum. Second, when the enterprise evolves to the balance trajectory of low exploratory innovation capability and high exploitative innovation capability, the willingness of the scientific research layer and the support layer to participate in the innovation balance is extremely low. The green manufacturing innovation ecosystem operates efficiently, but with low innovation efficiency, so the marginal cost of system operation gradually

decreases. However, when the enterprise focuses on exploitative innovation, it tends to degrade its ability to perceive and adapt to changes in the national or industry innovation environment, making it challenging to develop new products and services and fall into the "innovation trap." Third, when the enterprise evolves to a higher state of interactive innovation balance, they balance new technology development and existing technological innovation architecture. They can compensate for the total ecosystem gains by optimizing exploitative innovation before transforming the new technology. When exploitative innovation cannot sustain the evolution of the green manufacturing innovation ecosystem, the enterprise shifts its innovation resources to exploratory innovation and screens out some existing technologies and products. However, when brand-new technologies and products are launched, there is room for improvement and optimization. At this point, the enterprise carries out exploitative innovation, which no longer enhances the total revenue but accumulates resources for implementing the next stage of exploratory innovation. In this interactive innovation balance dynamic evolution, the enterprise innovation balance spirals up, the scientific research and support layers have strong participation, and the green manufacturing innovation ecosystem has robust and sustainable development capability.

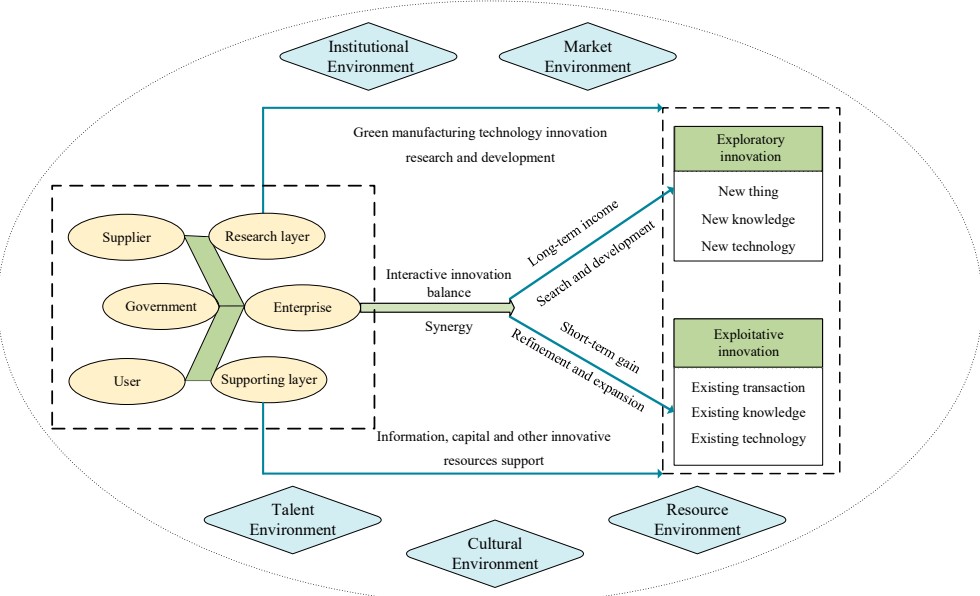

**Figure 1.** Interactive innovation balance structure of enterprises in green manufacturing innovation ecosystem.

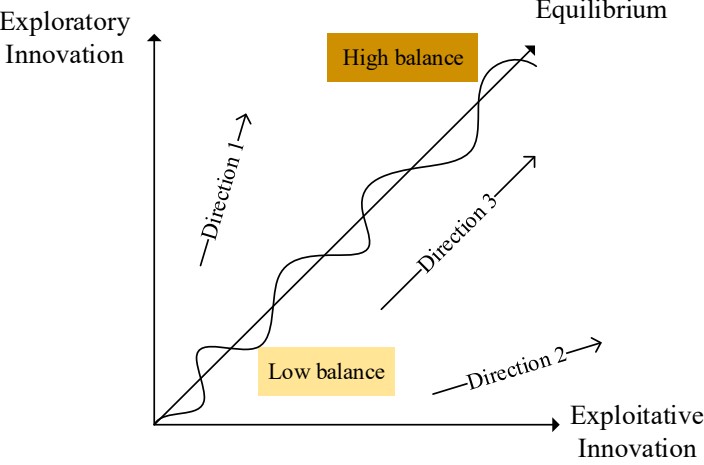

**Figure 2.** Trajectory of the evolutionary direction of interactive innovation balance.

## 4. Research Methods

### 4.1. Method Introduction and Hypothesis Development

The research perspective of the current literature is usually a specific research field or a disciplinary perspective, which has been explored from the local to overall level, effectively promoting the development of exploratory and exploitative innovation theory of enterprises. However, the following gaps remain. First, the research subject is relatively single and needs a solution unlike traditional game theory. Evolutionary game theory does not presuppose that people are rational and require perfect information. Evolutionary game theory has its roots in biological evolution. Usually, it takes the population of participants as the object of study, reacting to the dynamic equilibrium in which all participants in the continuous strategy space transform from one strategy to another [40]. The evolutionary game theory focuses on synergistic cooperation between participants and is sometimes reduced to the evolution of cooperation. Nowadays, evolutionary game theory is widely used in academia to solve significant scientific problems, such as complex interactions [41], circular dynamics [42], and multi-participant interactions [43], and also to analyze more complex social issues [44]. Further, we choose the evolutionary game approach for our analysis and make the following assumptions.

**Hypothesis 1.** *In the innovation balance process of the enterprise under the green manufacturing innovation ecosystem, the strategy adopted by the enterprise is (engage, no engage) with probability$(x, 1 - x)$. The strategy adopted by the support layer is (support, no support) with probability$(y, 1 - y)$. The strategy adopted by the scientific research layer is (collaborate, no collaborate) with probability $(z, 1 - z)$.*

**Hypothesis 2.** *"Collaborate" refers to the use of significant resources and energy at the research level for green manufacturing technology exploration innovation, taking into account utilization-based innovation and promoting a balance between the two; at the same time, "conflict" occurs for companies that rely only on utilization-based innovation, reducing the R&D support for green manufacturing technology innovation. "No collaborate" refers to the fact that the scientific research layer, in collaboration with the enterprise and the support layer, takes and expands existing innovation capabilities, technologies, and paradigms to maintain the development of the enterprise's utilization-based innovation but does not promote the balanced evolution of enterprise innovation. The degree of support from the scientific research layer to the enterprise and the support layer varies considerably under different exploratory innovation maturity levels. In addition, when the support layer actively supports the interactive innovation balance, it delivers information resources such as practical market information and user feedback for the enterprise and the scientific research layer.*

**Hypothesis 3.** *Based on the division of technology span degree, green manufacturing technology innovation balance activities can be divided into exploitative innovation and exploratory innovation according to technology span. Drawing on the research results of Sheng et al. [45], a comprehensive technology degree was selected to measure technological innovation balance activities. Technical innovation balance activities were divided into technology maturity and the technological innovation degree. Technology maturity indicates the maturity of current technology development. It is the ratio of enterprises using the technology to the total number of enterprises in the same industry, denoted by $T$. High technology maturity means that the enterprise favors exploitative innovation. The degree of technological innovation indicates the degree of innovation of current technology R&D. It is the ratio of the number of patents of this technology to the total number of enterprise patents, expressed by $\alpha$. The higher the degree of technological innovation, the smaller the value of $\alpha$, then the enterprise is more inclined to exploratory innovation. Therefore, the technological innovation balance composite degree is $T^\alpha$, $T, \alpha \in (0, 1)$. Let $\omega = T^\alpha$; the smaller the value of $\omega$, the stronger the innovation balance capability. The maturity of exploitative and exploratory innovation is $T^1$, $T^2$, and the innovation degree is $\varepsilon^1, \varepsilon^2$, which satisfies $T^1 > T^2$ and $\varepsilon^1 > \varepsilon^2$.*

**Hypothesis 4.** *Based on the classical assumptions of the $A - J$ model, the cost of participating in the interactive innovation equilibrium process is set as $C = \frac{1}{2}\beta\omega^2$, and $\beta$ is the integrated cost factor. Using b, e, and f to denote the combined cost coefficients of the enterprise, the research layer, and the support layer, respectively, the combined cost of the enterprise "engage" in the innovation equilibrium process is $C_E = \frac{1}{2}b\omega^2$, the combined cost of the scientific layer "collaborate" in the innovation equilibrium process is $C_G = \frac{1}{2}e\omega^2$, and the combined cost of the support layer "support" the innovation equilibrium process is $C_U = \frac{1}{2}f\omega^2$.*

**Hypothesis 5.** *If an enterprise adopts an interactive innovation balance strategy of "not engage", the existing green manufacturing innovation R&D system will produce stable benefits $R_0$, but the stability of coping with the impact of technological change is weakened, and the negative effect is $P_E$. If the scientific research layer adopts a "collaborate" strategy, it will provide the enterprise with technological innovation results continuously with an upper limit of $B_E$ and a benefit coefficient of $\alpha$, $\alpha \in (0,1)$. Meanwhile, the scientific research layer collects additional innovation R&D costs from enterprises, capped at a $F_E$ factor of $\chi$, $\chi \in (0,1)$.*

**Hypothesis 6.** *If the enterprise adopts the "engage" interactive innovation equilibrium strategy, the demand for the new knowledge and technology developed by the scientific research layer will continue to grow, and the scientific research layer can obtain the benefit of $D_G w$; if the enterprise adopts the " no engage" strategy, the demand for the new knowledge and technology will be weakened, and the scientific research layer will have "excessive R&D", and the additional resource loss is $P_G$. The overall benefits of the innovation ecosystem are enhanced if the scientific research layer and enterprises adopt the "non-participation" and "engagement" strategies, respectively, in which the spillover benefits that the scientific research layer will obtain from the green manufacturing innovation ecosystem are $R_G w$.*

**Hypothesis 7.** *If the support layer adopts a "support" strategy, it will actively search for customer needs in emerging markets and provide feedback to the enterprise, as well as deliver effective information to the scientific research layer. In the case of "collaborate" and "support" strategies at the scientific research layer and support layer, respectively, the scientific layer will reward the support layer with $B_U$. If the scientific research layer adopts a "no collaborate" strategy, it will lose part of the innovation R&D revenue, with an upper limit of $E_G$ and a factor of $\beta$, $\beta \in (0,1)$. If the company adopts a "no participate" strategy, it will reduce the frequency of access to information for the support layer significantly, and the direct loss of revenue for the support layer is $G_U$. If the enterprise and the support layer adopt the "engage" and "support" strategies, respectively, the perceived benefit to the support layer from the technology spillover from the green manufacturing innovation ecosystem is $Mw$.*

In summary, the game payment matrix of innovation subjects in interactive innovation balance can be derived, as shown in Table 1.

**Table 1.** Payment matrix of enterprise interactive innovation balance game.

| | | Scientific Research Layer | | | |
|---|---|---|---|---|---|
| | | Collaborate ($y$) | | No Collaborate ($1-y$) | |
| | | Support Layer | | Support Layer | |
| | | Support ($z$) | No Support ($1-z$) | Support ($z$) | No Support ($1-z$) |
| Enterprise | Engage ($x$) | $-\frac{1}{2}bw^2 + \alpha B_E + R_0$ $-\frac{1}{2}ew^2 + D_G w - \alpha B_E$ $-\frac{1}{2}fw^2 + Mw$ | $-\frac{1}{2}bw^2 + \alpha B_E + R_0$ $-\frac{1}{2}ew^2 + D_G w - \alpha B_E$ $0$ | $-\frac{1}{2}bw^2 + R_0$ $R_G w$ $-\frac{1}{2}fw^2 + Mw$ | $-\frac{1}{2}bw^2 + R_0$ $R_G w$ $0$ |
| | No engage ($1-x$) | $R_0 - \chi F_E - P_E$ $-\frac{1}{2}ew^2 + \chi F_E - P_G - B_U$ $-\frac{1}{2}fw^2 + B_U - G_U$ | $R_0 - \chi F_E - P_E$ $-\frac{1}{2}ew^2 + \chi F_E - P_G$ $-G_U$ | $R_0 - \chi F_E - P_E$ $jF_E - P_G - B_U - \beta E_G$ $-\frac{1}{2}fw^2 + B_U - G_U$ | $R_0 - P_E$ $-P_G$ $-G_U$ |

### 4.2. Model Construction and Solution

The expected return function for the enterprise choosing the "engage" strategy is shown in (1).

$$U_{E1} = (y-1)(z-1)(R_0 - \frac{bw^2}{2}) + yz(R_0 + \alpha B_E - \frac{bw^2}{2}) - z(y-1)(R_0 - \frac{bw^2}{2}) - y(z-1)(R_0 + \alpha B_E - \frac{bw^2}{2}) \quad (1)$$

The expected return function for the enterprise choosing the "no engage" strategy is shown in (2).

$$U_{E2} = y(z-1)(P_E - R_0 + \chi F_E) + z(y-1)(P_E - R_0 + \chi F_E) - (y-1)(z-1)(P_E - R_0) - yz(P_E - R_0 + \chi F) \quad (2)$$

The average expected return function of the enterprise is shown in (3).

$$\overline{U_E} = R_0 - P_E + xP_E - x\frac{bw^2}{2} - y\chi F_E - z\chi F_E + xy\chi F_E + xz\chi F_E + yz\chi F_E + xy\alpha B_E - xyz\chi F_E \quad (3)$$

The expected return function for the "collaborate" strategy chosen by the scientific research layer is shown in (4).

$$U_{G1} = z(x-1)(\frac{ew^2}{2} + B_U + P_G - \chi F_G) - xz(\frac{ew^2}{2} - D_G w + \alpha B_E) - (x-1)(z-1)(\frac{ew^2}{2} + P_G - \chi F_E) + x(z-1)(\frac{ew^2}{2} - D_G w + \alpha B_E) \quad (4)$$

The expected return function of the "no collaborate" strategy for the scientific research layer is shown in (5).

$$U_{G2} = z(x-1)(B_U + P_G + \chi F_E + \beta E_G) - (x-1)(z-1)P_G + xzR_G w - x(z-1)R_G w \quad (5)$$

The average expected return function of the scientific research layer is shown in (6).

$$\overline{U_G} = \begin{aligned} &y(z(x-1)(\frac{ew^2}{2} + B_U + P_G - \chi F_E) - xz(\frac{ew^2}{2} - D_G w + \alpha B_E) - (x-1)(z-1)(\frac{ew^2}{2} + P_G - \chi F_E) \\ &+x(z-1)(\frac{ew^2}{2} - D_G w + \alpha B_E) + (y-1)(P_G(x-1)(z-1) - z(x-1)(B_U + P_G + \chi F_E + \beta E_G) - xzR_G w + x(z-1)R_G w) \end{aligned} \quad (6)$$

The expected return function for the "support" strategy selected for the support layer is shown in (7).

$$U_{U1} = xy(Mw - \frac{fw^2}{2}) + y(x-1)(\frac{fw^2}{2} - B_U + G_U) - x(y-1)(Mw - \frac{fw^2}{2}) - (x-1)(y-1)(\frac{fw^2}{2} - B_U + G_U) \quad (7)$$

The expected return function for the "no support" strategy for the support layer is shown in (8).

$$U_{U2} = y(x-1)G_U - (x-1)(y-1)G_U \quad (8)$$

The average expected return function of the support layer is shown in (9).

$$\overline{U_U} = zB_U - G_U + xG_U - z\frac{fw^2}{2} - xzB_U + xzMw \quad (9)$$

### 4.3. Stability Analysis of the Replication Dynamic Equation

The system of replicated dynamic equations is constructed from (1)–(9) to obtain (10)–(12).

$$F(x) = \frac{dx}{dt} = -\frac{x(x-1)}{2}(-bw^2 + 2P_E + 2y\alpha B_E + 2y\chi F_E + 2z\chi F_E - 2yz\chi F_E) \quad (10)$$

$$F(y) = \frac{dy}{dt} = \frac{y(y-1)}{2}(\frac{ew^2}{2} - 2\chi F_E + 2x\alpha B_E + 2x\chi F_E - 2z\chi F_E - 2z\beta E_G - 2xD_G w + 2xR_G w + 2xz\chi F_E + 2xz\beta E_G) \quad (11)$$

$$F(z) = \frac{dz}{dt} = -\frac{z(z-1)}{2}(-fw^2 + 2xMw + 2B_U - 2xB_U) \quad (12)$$

4.3.1. Stability Analysis of the Enterprise Game Strategy

The derivative of $F(x)$ in the game replication dynamic equation (10) for the enterprise is calculated to obtain (13).

$$F\prime(x) = \frac{dF(x)}{dx} = \frac{(1-2x)}{2}(-bw^2 + 2P_E + 2y\alpha B_E + 2y\chi F_E + 2z\chi F_E - 2yz\chi F_E) \quad (13)$$

Letting $F(x) = 0$, we obtain:

(1) When $-bw^2 + 2P_E + 2y\alpha B_E + 2y\chi F_E + 2z\chi F_E - 2yz\chi F_E = 0$, take $y_0 = (-bw^2 + 2P_E + 2z\chi F_E)/(2\alpha B_E + 2\chi F_E - 2z\chi F_E)$. It is known that when $y = y_0$, the enterprise's interactive innovation balance strategy is steady state in the game system at this time, regardless of the value of $x$;

(2) When $F(x) = 0$ and $y \neq y_0$, it is obtained that $x_1 = 0$ and $x_2 = 1$ are two equilibrium state points in the evolution of the enterprise game.

Combined with the principle of stability of the differential equation, when $F(x) = 0$ and $F\prime(x) < 0$, the enterprise adopts the game strategy of "engage" in a stable state. In turn, the different cases of the value range of $-bw^2 + 2P_E + 2y\alpha B_E + 2y\chi F_E + 2z\chi F_E - 2yz\chi F_E$ are discussed. Letting $G(y) = -bw^2 + 2P_E + 2y\alpha B_E + 2y\chi F_E + 2z\chi F_E - 2yz\chi F_E$ gives $\frac{\partial G(y)}{\partial y} = 2\alpha B_E + 2(1-z)\chi F_E > 0$, then $G(y)$ is an increasing function with respect to $y$. Thus, when $y > y_0$, $G(y) > 0$, then $F\prime(1) < 0$, $F\prime(0) > 0$, $x = 1$ is the evolutionary state of the game for enterprise, and the enterprise tends to "participate" in the interactive innovation balance. When $y < y_0$, $G(y) < 0$, then $F\prime(1) > 0$, $F\prime(0) < 0$, and $x = 0$ is the evolutionary state of the game, then the enterprise tends to "no engage" in the interactive innovation balance.

The evolutionary process of enterprise decision-making behavior can be derived from the above analysis, as shown in Figure 3.

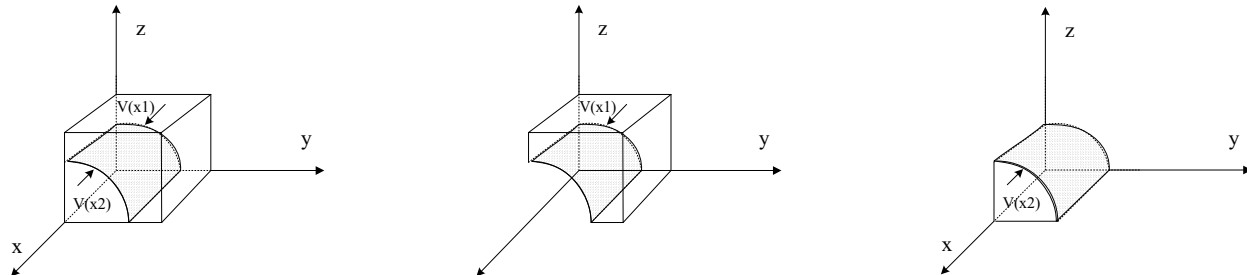

**Figure 3.** Evolutionary process of enterprise decision-making behavior.

**Conclusion 1.** *When the state of the strategy space chosen by the enterprise is located in the space $V(x_1)$, then $G(y) > 0$, $P_E + y\alpha B_E + (y + z - zy)\chi F_E > \frac{bw^2}{2}$ and $x = 1$ is the stable equilibrium point in the space $V(x_1)$. It means that when enterprises actively engage in the interactive innovation balance, the technological R&D support from the scientific research layer generates higher benefits than the integrated costs. Thus, the integrated benefits of the green manufacturing innovation ecosystem increase and are positively correlated with the innovation balance. As time evolves, the game strategy of enterprises eventually stabilizes by adopting the intermittent innovation balance of "engage".*

**Conclusion 2.** *If the enterprise chooses a strategic space state located in the space $V(x_2)$, then $G(y) < 0$, $P_E + y\alpha B_E + (y + z - zy)\chi F_E < \frac{bw^2}{2}$, and $x = 0$ is the stable equilibrium point in the space $V(x_2)$. This means that the cost of actively engaging in the interactive innovation balance is greater than traditional innovation and even higher than the negative benefits of "no engage" in the innovation balance. As time evolves, the enterprises' game strategy eventually stabilizes by adopting the "no engage" innovation balance process.*

According to Figure 3, further analysis of the gain in enterprise decision-making behavior shows that when $\chi$ increases and other parameters remain constant, the value of $y_0$ becomes prominent, the cross-section shifts downward, and $V(x_1)$ increases. This indicates that the R&D support at the scientific research level positively promotes the enterprise to engage in interactive innovation balance. While when $b$ increases and other parameters remain unchanged, the value of $y_0$ becomes smaller and the cross-section shifts upward, indicating that the input cost of the enterprise to engage in innovation balance gradually becomes larger and larger than the comprehensive benefit, and the final game strategy of enterprises tends to "no engage".

### 4.3.2. Stability Analysis of the Scientific Research Layer Game Strategy

The derivative of $F(y)$ in the replication dynamic equation (11) for the scientific research layer game is calculated to obtain (14).

$$F\prime(y) = \frac{\partial F(y)}{\partial y} = \frac{2y-1}{2}\left(\frac{ew^2}{2} - 2\chi F_E + 2x\alpha B_E + 2x\chi F_E - 2z\chi F_E - 2z\beta E_G - 2xD_G w + 2xR_G w + 2xz\chi F_E + 2xz\beta E_G\right) \quad (14)$$

Letting $F(y) = 0$, we obtain:

(1) When $\frac{ew^2}{2} - 2\chi F_E + 2x\alpha B_E + 2x\chi F_E - 2z\chi F_E - 2z\beta E_G - 2xD_G w + 2xR_G w + 2xz\chi F_E + 2xz\beta E_G = 0$, take $x_0 = \left(\frac{ew^2}{2} - 2\chi F_E - 2z\chi F_E - 2z\beta E_G\right)/(2\alpha B_E + 2\chi F_E - 2D_G w + 2R_G w + 2z\chi F_E + 2z\beta E_G)$, it is known that $F(y) = 0$ when $x = x_0$. At this point, the innovation balance strategy of the scientific research layer is steady state, regardless of the value of $y$ taken.

(2) When $x \neq x_0$ and $F(y) = 0$, we can obtain $y_1 = 0$ and $y_2 = 1$ as two equilibrium stable points. Combined with the principle of stability of the differential equation, when $F(y) = 0$ and $F\prime(y) < 0$ conditions hold, the scientific research layer takes the "collaborate" strategy for the stable state.

Further, the range of values of $\frac{ew^2}{2} + (-2 + 2x - 2z + 2xz)\chi F_E + 2x\alpha B_E + (-2z + 2xz)\beta E_G - 2xD_G w + 2xR_G w$ is discussed. Let $H(x) = \frac{ew^2}{2} + (-2 + 2x - 2z + 2xz)\chi F_E + 2x\alpha B_E + (-2z + 2xz)\beta E_G - 2xD_G w + 2xR_G w$ and find $\frac{\partial H(x)}{\partial x} = 2\alpha B_E + 2(1 + z)\chi F_E - 2D_G w + 2R_G w + 2z\beta E_G > 0$, then $H(x)$ is an increasing function about $x$.

(3) When $x > x_0$, $H(x) > 0$, then $F\prime(0) < 0$, $F\prime(1) > 0$, then $y = 0$ is the stable state of the game evolution of the scientific research layer, so the scientific research layer tends to reduce the collaboration with the enterprise in green manufacturing technology innovation R&D, that is, the strategy is stable in "no collaborate".

(4) When $x < x_0$, $H(x) < 0$, then $F\prime(0) > 0$ and $F\prime(1) < 0$, then $y = 1$ is the stable state of the game evolution of the scientific research layer. Therefore, the scientific research layer tends to actively research and develop green manufacturing technology innovation for the enterprise, and the strategy is stable in "collaborate".

The evolutionary process of decision-making behavior at the scientific research level can be obtained from the above analysis, as shown in Figure 4.

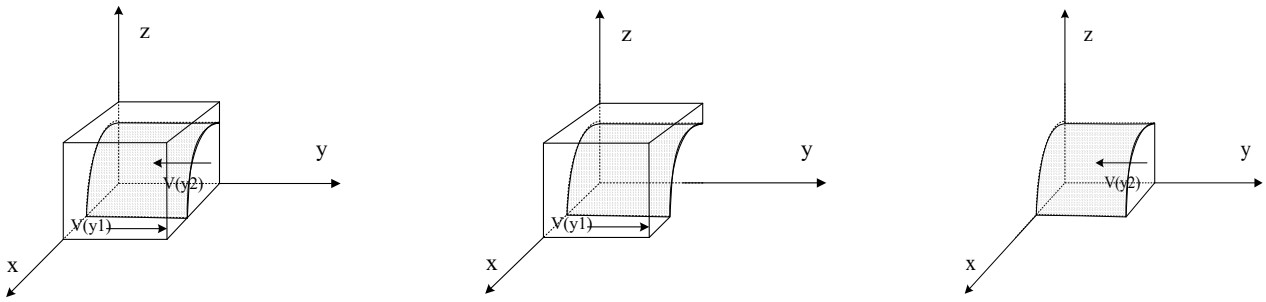

**Figure 4.** Evolutionary process of the scientific research layer decision-making behavior.

**Conclusion 3.** *When the initial state of the scientific research layer selection strategy is located in the space $V(y_1)$, $H(x) > 0$, then $(-2 + 2x - 2z + 2xz)\chi F_E + 2x\alpha B_E + (-2z + 2xz)\beta E_G + 2xR_Gw > 2xD_Gw - \frac{ew^2}{2}$, and at this point $y = 1$ is the stable equilibrium point of the space $V(y_1)$. The implication is that the adoption of "collaborate" strategy by the scientific research layer will bring more comprehensive positive benefits to the green manufacturing innovation ecosystem than traditional innovation activities. Therefore, as the innovation balance evolves, the scientific research layer's game strategy stabilizes at "collaborate".*

**Conclusion 4.** *When the initial state of the strategy chosen by the scientific research layer is located in the space $V(y_2)$, $H(x) < 0$, $(-2 + 2x - 2z + 2xz)\chi F_E + 2x\alpha B_E + (-2z + 2xz)\beta E_G + 2xR_Gw > 2xD_Gw < \frac{ew^2}{2}$ and $y = 0$ is the stable equilibrium point in the space $V(y_2)$. The implication is that adopting a "collaborate" strategy by the scientific research layer imposes high costs on the green manufacturing innovation ecosystem, which exceed the combined benefits. Therefore, as the innovation balance evolves, the game strategy of the scientific research layer stabilizes at "no collaborate".*

According to Figure 4, further analysis of the gain in the decision-making behavior of the scientific research layer shows that when $F_E$ increases and other parameters remain unchanged, $x_0$ decreases, the interface shifts down, $V(y_1)$ becomes prominent, and $V(y_2)$ becomes smaller, indicating that the scientific research layer adopts a "collaborate" strategy to charge high royalties for not engaging in the interactive innovation balance strategy, which in turn brings high benefits to the scientific research layer, so the decision-making behavior of the scientific research layer gradually tends to be "collaborate". When the value of e increases, $x_0$ becomes larger, the interface moves up, $V(y_1)$ becomes smaller, and $V(y_2)$ becomes larger, indicating that the scientific research layer pays more for the "collaborate" strategy. Its game strategy tends to adopt "no collaborate".

### 4.3.3. Stability Analysis of the Support Layer Game Strategy

The derivative of $F(Z)$ in the analysis of the replication dynamic equation (12) for the support layer game is calculated to obtain (15).

$$F(z) = \frac{\partial F(z)}{dz} = \frac{(1-2z)}{2}(-fw^2 + 2xMw + 2B_U - 2xB_U)$$ (15)

Letting $F(z) = 0$, we obtain:

(1) When $-fw^2 + 2xMw + 2B_U - 2xB_U = 0$, taking $x_0 = -fw^2 + 2xMw + 2B_U - 2xB_U$, it is known that $F(z) = 0$ when $x = x_1$, at which time the support layer game strategy is stable in the system, regardless of the value of $z$.

(2) When $x \neq x_1$ and $F(z) = 0$, it is known that $x_1 = 0$ and $x_2 = 1$ are the two equilibrium points of the game strategy of the support layer. Combined with the principle of stability of differential equation, when $F(z) = 0$ and $F\prime(z) < 0$, the game strategy of "support" of the support layer is in a stable state.

Further, the range of values of $-fw^2 + 2xMw + 2B_U - 2xB_U$ in different cases is discussed.

Letting $J(x) = -fw^2 + 2xMw + 2B_U - 2xB_U$ gives $\frac{\partial J(x)}{\partial x} = 2Mw - 2B_U$, and if $Mw < B_U$, then $\frac{\partial J(x)}{\partial x} < 0$ and $J(x)$ is a decreasing function about $x$. If $Mw > B_U$, then $\frac{\partial J(x)}{\partial x} > 0$ and $J(x)$ is an increasing function with respect to $x$.

When $Mw < B_U$:

(3) When $x > x_1$, $J(x) < 0$, then $F\prime(0) < 0$, $F\prime(1) > 0$, at this point $z = 0$ is the stable state of the evolutionary game of the support layer. Therefore, the support layer does not tend to provide information, capital, and other innovation resources to support the enterprise, and the game strategy is stable at "no support".

(4) When $x < x_1$, $J(x) > 0$, then $F\prime(0) > 0$, $F\prime(1) < 0$, at this point $z = 1$ is the stable state of the evolutionary game of the support layer. Therefore, the support layer tends

to provide information, capital, and other innovation resources to support enterprises, and the game strategy is stable at "support".

When $Mw > B_U$:

(5) When $x > x_1$, $J(x) > 0$, then $F\prime(0) > 0$, $F\prime(1) < 0$, at this point $z = 1$ is the stable state of the evolutionary game of the support layer. Therefore, the support layer tends to provide information, capital, and other innovation resources to support enterprises, and the game strategy is stable at "support".

(6) When $x < x_1$, $J(x) < 0$, then $F\prime(0) < 0$, $F\prime(1) > 0$, at this point $z = 0$ is the stable state of the evolutionary game of the support layer. Therefore, the support layer does not tend to provide information, capital, and other innovation resources to support the enterprise, and the game strategy is stable at "no support".

From the above analysis, the evolutionary process of the decision-making behavior of the support layer can be obtained, as shown in Figure 5.

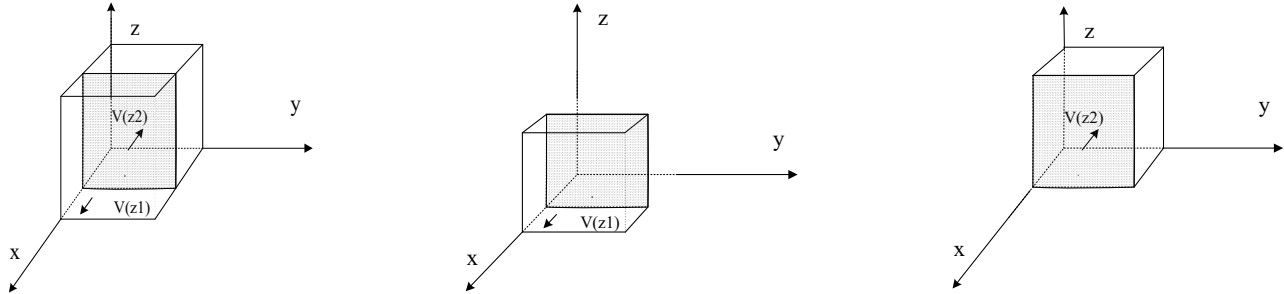

**Figure 5.** Evolutionary process of the support layer decision-making behavior.

**Conclusion 5.** *When the initial state of the support layer selection strategy is located in the space $V(z_1)$, $J(x) > 0$, $2xMw + 2(1 - x)B_U > fw^2$, then $z = 1$ is the stable equilibrium point in the space $V(z_1)$. At this point, the reward and perceived benefit of the research layer is higher than the total cost of the "support" decision. As time evolves, the game state of the support layer finally stabilizes at "support".*

**Conclusion 6.** *When the initial state of the support layer selection strategy is located in the space $V(z_2)$, $J(x) < 0$, $2xMw + 2(1 - x)B_U < fw^2$, then $z = 0$ is a stable equilibrium point in the space $V(z_2)$. At this point, the cost of providing innovation resources such as information and capital by the support layer is higher than the reward and perceived benefit obtained. As time evolves, the game state of the support layer eventually stabilizes at "no support".*

According to Figure 5, further analysis of the gain in decision-making behavior of the support layer shows that when $B_U$ or $M$ increases and other parameters remain unchanged, $x_0$ increases, the cross-section shifts down, $V(z_1)$ increases, and $V(z_2)$ decreases, indicating that the support layer "supports" participation in the interactive innovation balance, and the rewards and perceived benefits are higher than the costs paid, and its decision-making behavior tends to "support".

*4.4. Portfolio Strategy Stability Analysis*

The system's evolutionary stability strategy (*ESS*) can be obtained by the local stability analysis of the Jacobi matrix of the system, and the Jacobi matrix *J* can be obtained according to the calculation of the system of replica dynamic equations. Since the calculation results of the Jacobi matrix are complicated, they are not shown in detail in the paper. According to the

evolutionary game theory, the equilibrium point that satisfies all non-positive eigenvalues of the Jacobi matrix is the evolutionary stability point of the system.

$$J = \begin{pmatrix} \frac{dF(x)}{dx} & \frac{dF(x)}{dy} & \frac{dF(x)}{dz} \\ \frac{dF(y)}{dx} & \frac{dF(y)}{dy} & \frac{dF(y)}{dz} \\ \frac{dF(z)}{dx} & \frac{dF(z)}{dy} & \frac{dF(z)}{dz} \end{pmatrix} = \begin{pmatrix} J_1 & J_2 & J_3 \\ J_4 & J_5 & J_6 \\ J_7 & J_8 & J_9 \end{pmatrix} \quad (16)$$

The determinant and trace of the Jacobi matrix $J$ at the eight local equilibrium points are calculated as shown in Table 2.

**Table 2.** Eigenvalues of Jacobi matrix.

| Balance Points | Eigenvalue $\lambda_1$ | Eigenvalue $\lambda_2$ | Eigenvalue $\lambda_3$ | Stability |
|---|---|---|---|---|
| (0,0,0) | $P_E - \frac{bw^2}{2}$ | $\chi F_E - \frac{ew^2}{2}$ | $B_U - \frac{fw^2}{2}$ | Saddle Point |
| (0,1,0) | $P_E + \alpha B_E + \chi F_E - \frac{bw^2}{2}$ | $\frac{ew^2}{2} - \chi F_E$ | $B_U - \frac{fw^2}{2}$ | Instability point |
| (0,0,1) | $P_E + \chi F_E - \frac{bw^2}{2}$ | $2\chi F_E + \beta E_G - \frac{ew^2}{2}$ | $\frac{fw^2}{2} - B_U$ | Instability point |
| (0,1,1) | $P_E + \alpha B_E + \chi F_E - \frac{bw^2}{2}$ | $\frac{ew^2}{2} - 2\chi F_E - \beta E_G$ | $\frac{fw^2}{2} - B_U$ | Saddle Point |
| (1,0,0) | $\frac{bw^2}{2} - P_E$ | $D_G w - \alpha B_E - R_G w - \frac{ew^2}{2}$ | $Mw - \frac{fw^2}{2}$ | Instability point |
| (1,1,0) | $\frac{bw^2}{2} - P_E - \alpha B_E - \chi F_E$ | $\alpha B_E - D_G w + R_G w + \frac{ew^2}{2}$ | $Mw - \frac{fw^2}{2}$ | Instability point |
| (1,0,1) | $\frac{bw^2}{2} - P_E - \chi F_E$ | $D_G w - \alpha B_E - R_G w - \frac{ew^2}{2}$ | $\frac{fw^2}{2} - Mw$ | Saddle Point |
| (1,1,1) | $\frac{bw^2}{2} - P_E - \alpha B_E - \chi F_E$ | $\alpha B_E - D_G w + R_G w + \frac{ew^2}{2}$ | $\frac{fw^2}{2} - Mw$ | Stable point |
| $(x*,y*,z*)$ | | Saddle Point | | |

According to the discrimination principle of the local stability of the Jacobian matrix, it can be seen that when $del(J) > 0$ and $tr < 0$, the equilibrium point is the system evolution equilibrium point. We discuss the stability of local equilibrium points in different situations by combining them with BYD's innovation and evolution.

**Scenario 1.** *When $\frac{bw^2}{2} < P_E + \chi F_E$, $\alpha B_E + R_G w + \frac{ew^2}{2} > D_G w$, $Mw > \frac{fw^2}{2}$, the cost and benefit of traditional innovation by the innovation balance participants are smaller than the cost of innovation balance, which is in line with the high-cost characteristics of exploratory and exploitative innovation in green manufacturing in reality. In turn, the green manufacturing innovation ecosystem tends to be in the local stable state of "engage, no collaborate, support", and the evolutionary equilibrium point is (1,0,1).*

**Scenario 2.** *When $P_E + \alpha B_E + \chi F_E > \frac{bw^2}{2}$, $D_G w > R_G w + \frac{ew^2}{2} + \alpha B_E$, $Mw > \frac{fw^2}{2}$, the benefits of innovation balance are greater than the costs, and the green manufacturing innovation ecosystem tends to the local stable state of " engage, collaborate, support ", and the evolutionary equilibrium point is (1,1,1).*

**Scenario 3.** *When $\frac{bw^2}{2} > P_E + \alpha B_E + \chi F_E$, $2\chi F_E + \beta E_G > \frac{ew^2}{2}$, $B_U > \frac{fw^2}{2}$, the comprehensive cost of "engage" in the innovation balance strategy is greater than the sum of the negative benefits and R&D support. In turn, the cost of engaging in innovation balance activities is more prominent, harming the enterprise's established green manufacturing technology innovation system. Therefore, the game tends to a local stable state of "no engage, collaborate, support ", and the evolutionary equilibrium point is (0,1,1).*

## 5. Numerical Simulation

Case studies effectively solve most of today's scientific problems [46]. The analysis process usually has no specific criteria for the research sample. However, the internal logic should be consistent with the research question. Previous research has shown that case studies have been widely used in innovation ecosystems regarding core subject synergy and technological model innovation. Placing the interactive innovation balance in the

green manufacturing innovation ecosystem, we consider the impact of the decision-making behavior of the scientific research layer and support layer on balance between exploratory and exploitative innovation in the enterprise. BYD, a benchmark enterprise in the domestic new energy vehicle industry, has been making exploratory innovations in green manufacturing technologies in recent years. BYD has produced many exploratory innovations in core technologies such as the battery, motor, and electric control and has built an open innovation ecosystem to accelerate the ecological development of the whole innovation, industrial, and value chains [47]. In the green manufacturing innovation ecosystem with BYD enterprise as the core, innovation subjects actively participate, such as universities, research institutes, and other research layers, and collaborate with BYD for technology R&D. The support layer actively feeds external information such as user experience to BYD and makes a financial investment. Thus, the innovation subjects collaborate to enhance the sustainable competitiveness of BYD's new energy vehicle innovation ecosystem [48].

Based on considering the reality of BYD's green manufacturing innovation ecosystem, the evolutionary game theory analyzes the influence mechanism of the maturity and innovation degree of enterprise technological innovation, technological R&D at the scientific research layer, and the support of innovation resources such as information and capital provided at the support layer on the balance of interactive innovation of enterprises. It verifies the direct correlation of the evolutionary process of innovation game subjects. In order to further clarify the influence of the model's hypothetical variables, the trajectory, and the stability of the evolution of enterprise innovation balance, a case study is used, and the decision model is validated by simulation. The parameter assignments are shown in Table 3.

**Table 3.** Model parameters and assignments.

| Parameters | Assignment | Parameters | Assignment | Parameters | Assignment |
|:---:|:---:|:---:|:---:|:---:|:---:|
| $b$ | 80 | $e$ | 48 | $f$ | 16 |
| $w$ | 0.5 | $\alpha$ | 0.25 | $\beta$ | 0.2 |
| $B_E$ | 20 | $P_E$ | 2 | $F_E$ | 12.5 |
| $\chi$ | 0.8 | $D_G$ | 45 | $R_G$ | 30 |
| $B_U$ | 3 | $E_G$ | 5 | $M$ | 6 |

*5.1. The Impact of the Initial Willingness of Innovation Subjects on the Enterprise Innovation Balance*

Initial willingness is a subjective indicator of game subjects under rational decision-making. In order to analyze the influence of the initial willingness of innovation game subjects on the innovation balance of enterprises to reach the equilibrium state, four initial differentiated willingness values of (0.2,0.4,0.6,0.8) are set, and the dynamic evolution of innovation game subjects are plotted, respectively. As seen in Figure 6, when the benefits of participation in interactive innovation balance are higher than the costs for the enterprise, the scientific research layer, and the support layer, the initial willingness does not affect the steady-state direction of the innovation ecosystem. However, it only changes the rate of convergence to a steady state. When the initial cooperation willingness of the scientific research layer and the enterprise increases, the rate of their convergence to participate in the steady state of the innovation balance strategy becomes faster. Differently, the evolutionary process of the support layer is more sensitive to the change of the initial willingness and tends to the steady state at the lowest rate.

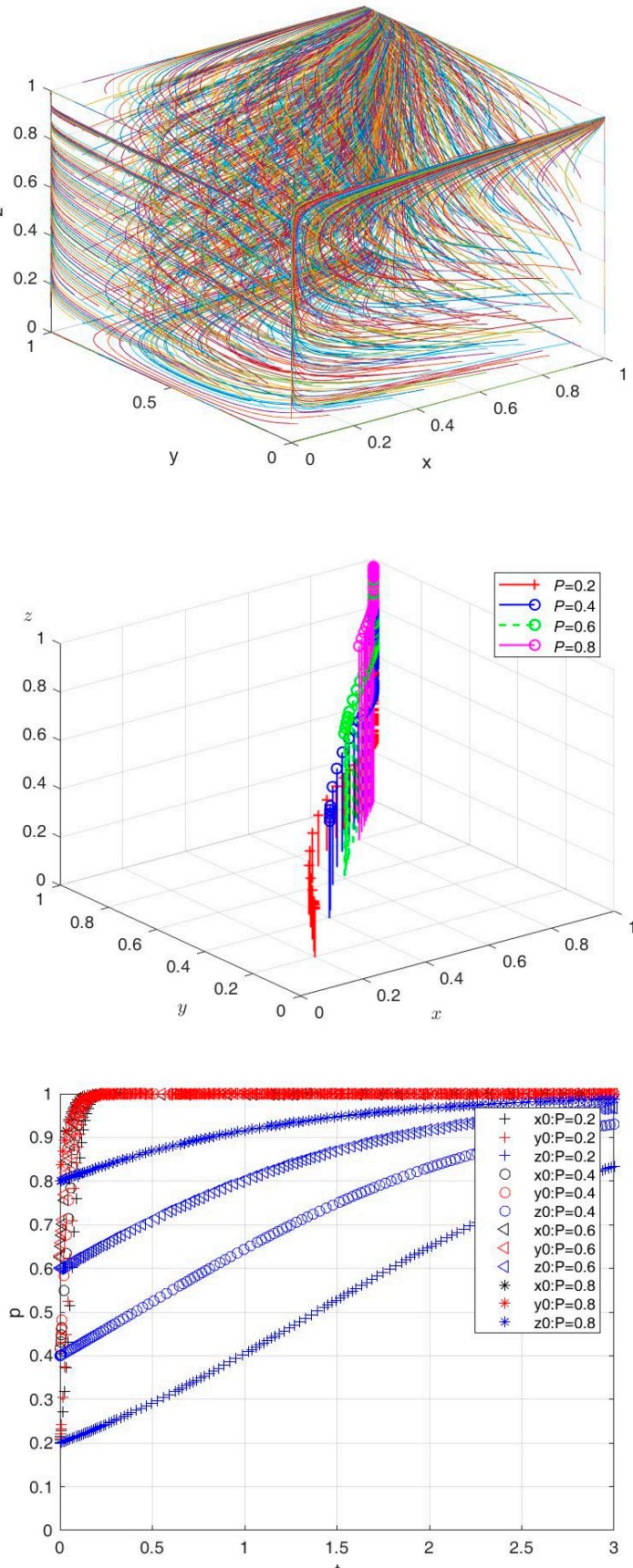

**Figure 6.** Impact of willingness to innovate on the evolutionary equilibrium of decision making.

## 5.2. The Impact of the Change of Innovation Balance Synthesis on the Enterprise Innovation Balance

The innovation balance composite degree is an intuitive indicator to quantify the interactive innovation balance of enterprises in the context of the green manufacturing innovation ecosystem. The values of the innovation balance synthesis degree are set as (0.2,0.4,0.6,0.8), and the influence of different innovation balance synthesis degrees on the evolution trend of innovation subjects is analyzed. As seen in Figure 7, when the innovation balance synthesis degree is lower than 0.4, the equilibrium state of the innovation balance system will not change. As the innovation balance synthesis degree gradually increases, the rate of the innovation subject tends to a steady state faster. However, when the innovation balance is high ($w = 0.8$), the support layer's decision-making behavior tends to be "no support". In turn, when the innovation balance is in the middle and high level ($w \geq 0.4$), the "complementary" behavior of the innovation subjects is apparent, and the enterprises actively engage in exploratory innovation to promote the innovation balance, which enhances the benefits of the scientific research layer and the support layer. When the innovation balance is at a low level ($w = 0.2$), the development of exploratory innovation is more mature, and the enterprise innovation balance is in a good state. However, the overall gain of the green manufacturing innovation ecosystem decreases.

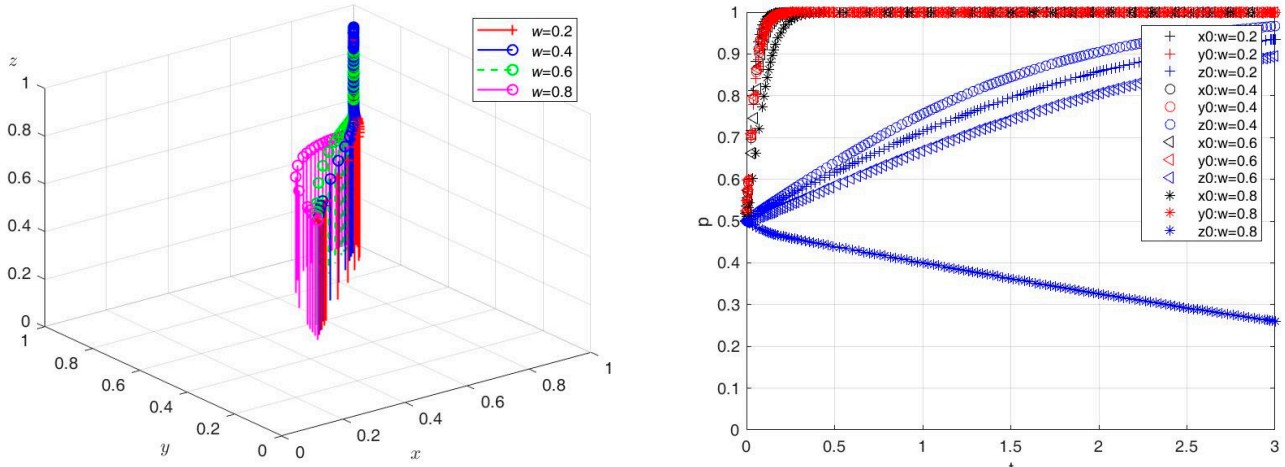

**Figure 7.** Impact of enterprise innovation balance changes on the evolutionary equilibrium of decision making.

## 5.3. The Impact of the Scientific Research Layer's Technology Development Efforts on the Enterprise Innovation Balance

The scientific research layer efforts to conduct green manufacturing technology R&D will contribute to the interactive innovation balance process of the enterprise. The coefficient of effort degree was set as (0.2,0.4,0.6,0.8) to obtain the differential influence of the scientific research layer's technology R&D decisions on the evolution of enterprise innovation balance. As seen in Figure 8, the level of effort does not change the steady state of the three-game subjects but only changes the rate at which the enterprise and the scientific research layer converge to the steady state. The higher the level of effort, the faster the rate at which the enterprise and the scientific research layer converge to the steady state, but the support layer's decision-making behavior is unaffected. Further, the scientific research layer's R&D of green manufacturing technology innovation for enterprises promotes the interactive type of equilibrium between exploratory and exploitative innovation of enterprises, and the sensitivity of enterprises to the level of effort is higher.

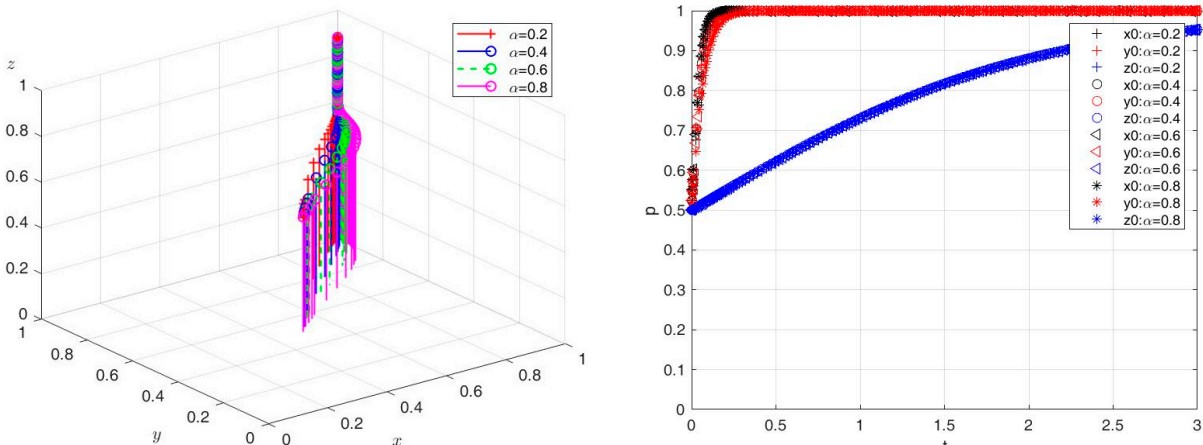

**Figure 8.** Impact of scientific research layer's technology R&D effort on the evolutionary equilibrium of decision making.

### 5.4. The Impact of the Level of "Conflict" between the Scientific Research Layer and the Enterprise on the Enterprise Innovation Balance

The "conflict" is not an absolute division between the enterprise and the scientific research layer, but on the contrary, it can promote positive decisions for both. Accordingly, the "no collaborate" strategy of the scientific research layer toward the enterprise reduces the sharing of green manufacturing technology results. It inhibits the sustainability of the green manufacturing innovation ecosystem. The maximum loss of the enterprise is set as $F_E = 12.5$, and the coefficients of "conflict" are set as (0.2,0.4,0.6,0.8) in order to consider the influence of different degrees of "conflict" on the evolution of the decision-making of innovation subjects. As seen in Figure 9, the degree of "conflict" changes the enterprise's homeostasis rate and the scientific research layer. However, it does not affect the decision-making of the support level. The decision-making behavior of the scientific research layer is more sensitive to the coefficient of "conflict", and as the coefficient increases, the willingness of the scientific research layer to "collaborate" becomes more muscular.

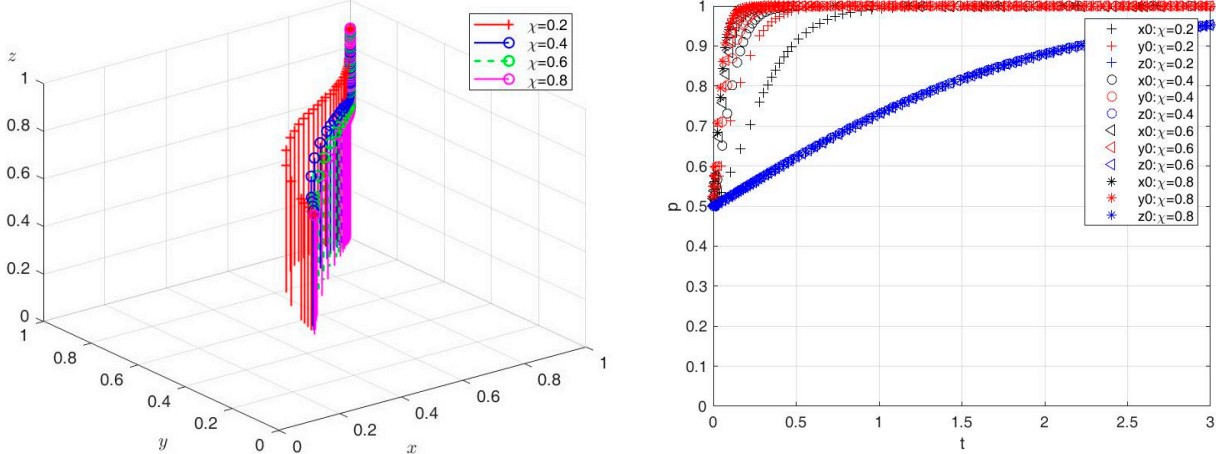

**Figure 9.** Impact of the degree of "conflict" between scientific research layer and enterprises on the evolutionary equilibrium of decision making.

### 5.5. The Impact of Changes in Perceived Benefits of the Support Layer on the Enterprise Innovation Balance

The perceived benefits at the support level imply positive effects from the interactive innovation balance process of the enterprise, setting the perceived benefits as (2,4,6,8). As

seen from Figure 10, the change in perceived benefits of the support layer does not change the steady-state changes of the enterprise and the scientific research layer but only affects its decision-making behavior. When the perceived return is low ($M = 2$), the "collaborate" probability of the support layer converges to 0, indicating that the low perceived return will prompt the support layer to carry out traditional innovation resource support. The support layer does not make decisions when the perceived benefit is at an intermediate level ($M = 4$). When the perceived benefit is at a high level ($M = 6$), the probability of "collaborate" converges to 1, implying that the higher perceived benefit will motivate the support layer to search for and provide resources such as information and funds for exploratory innovation.

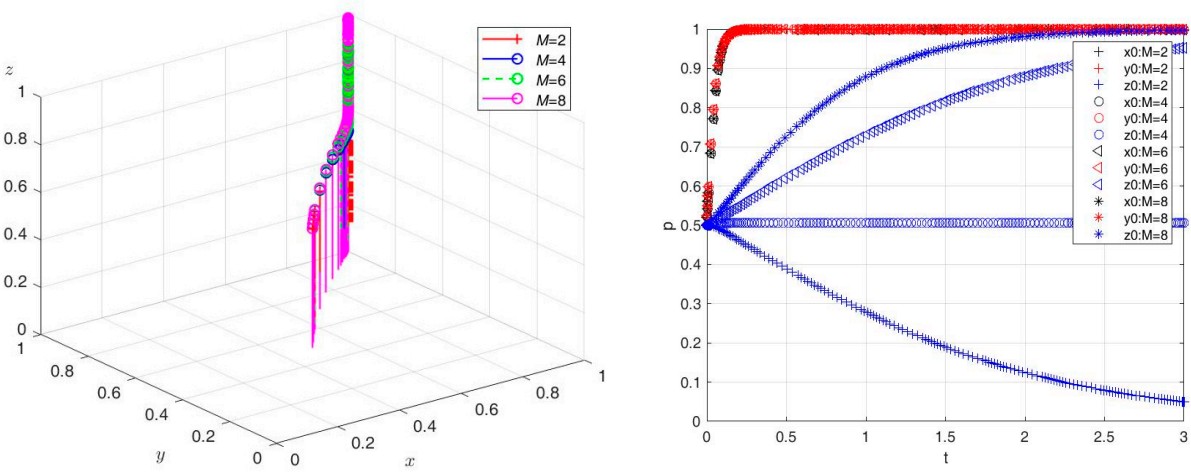

**Figure 10.** Effect of perceived benefits of support layer on decision evolution equilibrium.

## 6. Results

In this paper, we discuss the interactive innovation balance between exploratory and exploitative innovation of enterprises in the green manufacturing innovation ecosystem, build a complex network evolutionary game decision model involving three innovation subjects: enterprises, scientific research layer, and support layer, and systematically analyze the influence mechanism and inner law of decision-making behavior of innovation subjects. The research shows that (1) the balance between exploratory and exploitative innovation has direct feedback utility to the participating subjects. The higher the innovation balance, the more conducive it is to promote the interactive innovation balance of enterprises. The additional benefits generated by enterprises' active green manufacturing exploratory innovation to promote the innovation balance process are more significant than the costs paid. (2) The scientific research layer's R&D on green manufacturing technology innovation contributes to the exploratory innovation of the enterprise, which in turn is conducive to achieving innovation balance. The sensitivity of two decision-making behaviors, namely the sharing of green manufacturing technology results and the degree of "conflict" with the enterprise, led by the scientific research layer, is more evident in promoting the innovation balance of the enterprise. When the innovation balance increases, the willingness of the scientific research layer to participate increases significantly, which in turn enhances the overall benefit of the green manufacturing innovation ecosystem. (3) The degree of information, capital, and other innovation resources provided by the support layer to the green manufacturing innovation ecosystem has a positive effect on the interactive innovation balance of enterprises, in which the more prosperous the information resources searched from the outside, the more favorable the innovation balance of enterprises. The perceived benefits received by the support layer do not affect the decision-making of the enterprise and scientific research layer. However, they only have an impact on their decision-making behavior. Moreover, the support layer has a positive supervisory feedback

effect, which has a quality-enhancing effect on the evolution of the enterprise's decision-making and scientific research layer.

## 7. Discussions

In order to better promote the dynamic evolution of the interactive innovation balance between exploratory and exploitative innovation of enterprises and the sustainable development of the green manufacturing innovation ecosystem, this paper draws the following insights: first, improve the policy and institutional guarantee. The government can implement innovation incentives and financial support policies, set up special funds and tax incentives to enhance enterprises' exploratory innovation initiatives and regulate and maintain the existing utilization-based innovation capabilities and resources. Moreover, the government must encourage the active participation of other core subjects such as financial institutions, information intermediaries, and enterprises in the supply chain in the enterprise manufacturing innovation ecosystem to provide enterprises with technology, capital, information, and other security elements. Second, strengthen technical support. In the era of green manufacturing, the development of new infrastructures such as the Internet of Things, cloud computing, and big data should be coordinated to give full play to the empowering effect of new infrastructures for enterprises' green manufacturing exploratory innovation and their interactive innovation balance [49], break the core technology barriers, and enhance the green manufacturing technology innovation capability of enterprises' innovation balance from the newborn stage to the mature stage. Finally, strengthen the innovation environment support. Enterprises should create more opportunities for consumers to access new industries and services of innovation balance, continuously expand new markets, and discover potential customers. Meanwhile, enterprises should strengthen the connection with other innovation subjects to give full play to the complex network effect and technology accumulation effect of the green manufacturing innovation ecosystem and thus reduce the high cost of exploratory innovation.

Although this paper concludes that a higher innovation balance degree has a positive effect on the interactive innovation balance of firms and the sustainability of the green manufacturing innovation ecosystem, the adverse effects of a lower innovation balance degree were not examined. Therefore, future research can explore the relationship between different levels of innovation balance degrees and subject innovation balance, as well as the optimal degree interval to provide a practical reference for enterprise decision-making. In addition, the green manufacturing innovation ecosystem is a complex network structure that contains several heterogeneous subjects. However, this paper only considers the influential role of scientific research and support layers. Future research should consider the influence of other innovation agents, such as the influence of government regulation and subsidies, and try to quantify more factors.

**Author Contributions:** Conceptualization, H.Q. and H.J.; methodology, H.Q.; software, H.J.; validation, H.Q., H.Z. and H.J.; formal analysis, H.Q.; investigation, H.J.; resources, H.Z.; data curation, H.J.; writing—original draft preparation, H.Q.; writing—review and editing, H.Z.; visualization, H.J.; supervision, H.Q.; project administration, H.Z. All authors have read and agreed to the published version of the manuscript.

**Funding:** This research received no external funding.

**Institutional Review Board Statement:** Not applicable.

**Informed Consent Statement:** Not applicable.

**Data Availability Statement:** Data are available upon reasonable request.

**Acknowledgments:** The authors would like to thank the anonymous reviewers for their thoughtful suggestions and comments.

**Conflicts of Interest:** The authors declare no conflict of interest.

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
