# Peer review of "Research on Enterprise Interactive Innovation Balance Decision in Green Manufacturing Innovation Ecosystem"

_sustainability, doi:10.3390/su15107767_

Round 1

Reviewer 1 Report

The paper's topic and conducted research are very important and justified to be presented in a high-quality Journal. The subject is very important for the literature. The paper is informative, but some issues need to be addressed carefully. My decision is – a major revision, with some amendments. Please see my comments and suggestions below.

Comment 1. The authors should present the specific research questions in the Introduction.

Comment 2. Also, the authors should present research gaps and describe the work done in the Introduction. The paper can be a good example to help you improve your paper (Does Proactive Green Technology Innovation Improve Financial Performance? Evidence from Listed Companies with Semiconductor Concepts Stock in China. Sustainability 2022, 14, 4600. https://doi.org/10.3390/su14084600).

Comment 3. The title of 3.1 should change to “Interactive innovation balance structure”.

Comment 4. The process in Section 3 is confusing, for example, section 3.2.1. Basic Assumptions should not be placed in section 3. Model Construction.

Comment 5. The authors should indicate the used research methods in this study.

Comment 6. Maybe it is beneficial to introduce a section called: Results.

Comment 7. And discussion? The authors should be the discussions of the obtained results (author's original thoughts) in the light of the previous research. Moreover, section 6 should be included in the section discussions.

Comment 8. The language of this manuscript is bad and needs help from native speakers.

Good luck for your work!

Reviewer 2 Report

The paper seems to be very interesting. It shows decision making of triple helix actors acording the maturity of technological advanced and the balance between exploratory and exploitative technology eco-innovation. Although the technological maturity and learning by innovating are present in the contemporary literature. The decision-making behaviour related to support organization are kind of novelty to me. I'm a little bit surprised that authors used so few references from all around scientific world and focused mostly on Chinese scientists. The methods used are very sophisticated and the presentation of methods and model construction is low readable for the regular economic reader, I'm afraid. It would be good to rethink this part of the paper. Similar with all hypothesis which are very long and complicated.

Reviewer 3 Report

I am positive about this paper, where a theoretical model is tested through a proper simulation methodology. I would suggest two revisions:

1.      Some of the calculus should be moved to a dedicated appendix, since the reading might be difficult for a non-expert reader.

2.      Section 2.1 should be extended discussing Chinese peculiarities with regard to institutional and environmental factors which may affect innovation in general. In more detail, before focusing on green innovation, a new general paragraph about innovation in China should be added. Here below some references which should be discussed in this additional paragraph.

Cumming, D., Rui, O., & Wu, Y. P. (2016). Political instability, access to private debt, and innovation investment in China. Emerging Markets Review, 29, 68–81.

Fang, L. H., Lerner, J., & Wu, C. P. (2017). Intellectual property rights protection, ownership, and innovation: Evidence from China. Review of Financial Studies, 30(7), 2446–2477.

Guo, D., Guo, Y., & Jiang, K. (2016). Government-subsidized R&D and firm innovation: Evidence from China. Research Policy, 45(6), 1129–1144.

Liu, S., Du, J., Zhang, w. and X. Tian (2021).Opening the box of subsidies: which is more effective for innovation?. Eurasian Business Review, 11, 421–449

Rong, Z., Wu, X. K., & Boeing, P. (2017). The effect of institutional ownership on firm innovation: Evidence from Chinese listed firms. Research Policy, 46(9), 1533–1551.

Tian, X. L., Kou, G., & Zhang, W. K. (2020). Geographic distance, venture capital and technological performance: Evidence from Chinese enterprises. Technological Forecasting and Social Change, 158, 120155.

Zhang, M., Mohnen, P. (2022). R&D, innovation and firm survival in Chinese manufacturing, 2000–2006. Eurasian Business Review, 12, 59–95.

Zhao, S. K., Xu, B. D., & Zhang, W. Y. (2018). Government R&D subsidy policy in China: an empirical examination of effect, priority, and specifics. Technological Forecasting and Social Change, 135, 75–82.

Round 2

Reviewer 1 Report

I appreciate the authors' effort to improve the paper. After a carefully reviewing your revised manuscript, I am highly satisfied with the changes that you have made, and I have no more comments to offer. I can recommend the publication of this research. I wish you well in taking your research forward.

The paper is accepted in present form.

Reviewer 2 Report

The paper is now more readable but still tough to go thru the methodological part. Correction are sufficient to me and paper should be published in a presented form.